# Exploit Unlabeled Data on the Server! Federated Learning via Uncertainty-aware Ensemble Distillation and Self-Supervision

## Abstract

Federated Learning (FL) is a distributed machine learning paradigm that involves the cooperation of multiple clients to train a server model. In practice, it is hard to assume that each client possesses large-scale data or many clients are always available to participate in FL for the same round, which may lead to data deficiency. This deficiency degrades the entire learning process. To resolve this challenge, we propose a Federated learning with entropy-weighted ensemble Distillation and Self-supervised learning (FedDS). FedDS reliably deals with situations where not only the amount of data per client but also the number of clients is scarce. This advantage is achieved by leveraging the prevalent unlabeled data in the server. We demonstrate the effectiveness of FedDS on classification tasks for CIFAR-10/100 and PathMNIST. In CIFAR-10, our method shows the improvement over FedAVG by 12.54% in data deficient regime, and by 17.16% and 23.56% in more challenging scenarios of noisy label or Byzantine client cases, respectively.

## 1 Introduction

Federated Learning (FL) is a distributed machine learning paradigm that involves the cooperation of multiple clients to train a server model (McMahan et al., 2017). In FL, a server model is trained as follows: 1) the server distributes the current server model to clients, 2) each client independently trains each model downloaded from the server with their available local data and sends back the resultant model to the server, 3) the server updates the server model with the collected locally-trained models, and 4) repeat the steps. By collecting updated model parameters at the server instead of raw client data, FL can mitigate the personal information leakage.

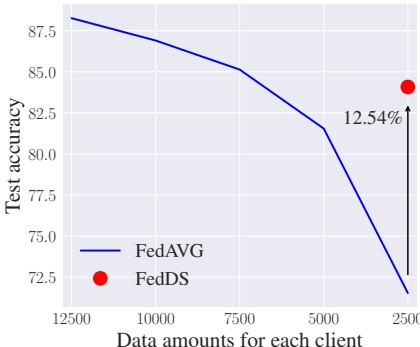

Figure 1: The accuracy of the standard FL method, FedAVG (McMahan et al., 2017), rapidly drops as the amount of available data at each client decreases. Our FedDS mitigates the effect of data deficiency by exploiting unlabeled data on the server. The accuracy is measured at the 50th communication round on CIFAR-10 classification task. This result follows the same setting as the main experiment in Fig. 4 except the data amounts for each client.

In some FL scenarios, such as developing a medical diagnosis algorithm, it is often the case where the number of clients (participating hospitals) and the size of labeled datasets in each client (the number of relevant patients and those labels at each hospital) are deficient. Such a lack of participating clients and labeled datasets in the clients leads to performance degradation for the standard FL method, *e.g.*, FedAVG (McMahan et al., 2017) (refer to Fig. 1). The deficiency may also destabilize the learning process which increases label noise sensitivity of FL methods. Even in such scenarios, unlabeled data is abundant or easy to collect in practice, which may help to mitigate the data deficiency and label noise vulnerability of FL algorithms.

In this paper, we propose a robust FL algorithm by utilizing additional unlabeled data on the server. The key idea of our method is to leverage unlabeled data to mitigate lack of data as well as to reliably

aggregate the client models into the single server model by unsupervised ensemble knowledge distillation. We postulate that the aforementioned degradation of the accuracy mainly stems from unreliable clients participated in the server model update. To mitigate the influence of unreliable clients, we measure the entropy of each client's output to assess the uncertainty of each model, and use them to weigh for each client model. We found that this simple entropy measure is sufficiently well-calibrated to train the server model better. Thereby, we suppress unreliable clients' contribution when aggregating them. Furthermore, our proposed entropy-weighted ensemble distillation (EED) is performed jointly with another self-supervised learning (SSL) loss with unlabeled data on the server. With our setting, while additionally imposing a SSL loss is simple and demands negligible overheads, it is found to be crucial and have several benefits, such as faster convergence and reducing the influence of unreliable nodes.

**Our major contributions**  We propose **Fed**erated learning with entropy-weighted ensemble **D**istillation and **S**elf-supervised learning (**FedDS**), a method for reliably updating the server model by utilizing the server's unlabeled data in an unsupervised way. We demonstrate through experiments that our FedDS outperforms several strong baselines in the classification tasks for CIFAR-10/100 and PathMNIST, especially in data deficient regime, and is also robust to various tough scenarios with unreliable clients.

## 2    RELATED WORK

Since the emergence of FL paradigm (McMahan et al., 2017), there have been many follow-up studies to tackle various challenges in FL: enhancing communication efficiency (Alistarh et al., 2017; Suresh et al., 2017; Bernstein et al., 2018; Tang et al., 2018; Wu et al., 2018; Hamer et al., 2020; Rothchild et al., 2020; Reisizadeh et al., 2020; Haddadpour et al., 2021; Qiao et al., 2021; Konečný et al., 2016; Hyeon-Woo et al., 2022; Jeong et al., 2018), stabilizing convergence and solving various issues arising from heterogeneity in client's data (Li et al., 2020; Karimireddy et al., 2020; Acar et al., 2021; Reddi et al., 2021; Yuan & Ma, 2020) and in client's model structure (Diao et al., 2021b), protecting clients from privacy attacks (Ammad-Ud-Din et al., 2019; Gong et al., 2021), and effectively aggregating unreliable client models containing malicious clients (Chang et al., 2019). In particular, the use of extra data, in addition to local data available at each client, has been shown to be effective to deal with some of the aforementioned challenges in FL (Chang et al., 2019; Lin et al., 2020; Li & Wang, 2019). In this section, we focus on briefly reviewing two main research streams utilizing additional data in FL, which are closely related to our work, *i.e.*, utilizing unlabeled data on the server. For comprehensive review of FL, please refer to Kairouz et al. (2019)

**Knowledge distillation in federated learning**  Knowledge distillation (KD) is used to transfer knowledge of a teacher model to a student model (Xie et al., 2020; Zhou et al., 2021; Radosavovic et al., 2018; Kim et al., 2021). The teacher model is used to provide pseudo labels to either labeled or unlabeled data, and the student model learns to mimic the teacher's behavior by using these pseudo-labeled data as supervision.

To adapt KD for FL, data to be pseudo labeled should be presented at the point of aggregation (the server); thereby, we can extract knowledge that will be transferred to the aggregated model (the server model). However, if the data to be distilled is provided from clients to the server, FL loses the privacy and security property which is one of the big advantages of FL. To address this, there are studies exploiting the assumptions that transferring features and logits from client's data is allowed (He et al., 2020; Gong et al., 2021) or assumptions that additional datasets exist (Gong et al., 2021; Chang et al., 2019; Lin et al., 2020; Li & Wang, 2019; Shi et al., 2021). The latter assumes that an extra public dataset is shared *across all clients*. Then, KD is performed at the server by collecting the logit values for the public dataset from the clients, instead of model parameters. This approach can achieve higher communication efficiency and privacy protection. Our work relaxes the assumption by possessing extra unlabeled data *only in the server*, rather than sharing across all clients.

As a similar setting to ours, Lin et al. (2020) proposes a FL method, called FedDF. In FedDF, the server obtains pseudo labels for the unlabeled data with the collected client models, and performs KD with those pseudo labels. They show that their approach is robust to non-i.i.d. data, achieving a higher accuracy than FedAVG (McMahan et al., 2017). We further take into account confidence of each client and core knowledge of the server by measuring uncertainty of client predictions and applying

self-supervised learning (SSL), respectively. Thereby, our proposed FedDS enhances both reliability and robustness of the learned model against the data deficiency and label noise vulnerability.

**Semi-supervised federated learning** Unlabeled data is much more prevalent than labeled ones. Semi-supervised learning approaches in FL (Shi et al., 2021; Jeong et al., 2021; Zhang et al., 2021; Diao et al., 2021a; Lin et al., 2021) have been studied to leverage these prevalent unlabeled data. All of these studies consider scenarios where each client has few labeled data but instead has a large amount of unlabeled data, and exploits the unlabeled data with unsupervised learning losses. Unlike these preceding studies, our work considers the scenario where unlabeled data is leveraged *on the server side* while clients have limited labeled data, which is a more practical setting.

## 3 METHOD

### 3.1 PROBLEM SETTING

We assume a set of $C$ clients with client index $c \in \{1, 2, ..., C\}$ and a single server. Each client with index $c$ has its own labeled dataset $\mathcal{D}_c = \{x_i, y_i\}_{i=1}^{N_c}$, where $(x_i, y_i) \in \mathcal{X} \times \mathcal{Y}$ for each $i$, $x_i$ is input data instance and $y_i$ is its corresponding label in the one-hot vector form. Client data distributions can be i.i.d. (each client has the same data distribution), or non-i.i.d. (each client has different data distribution with others). In our setting, the central server has an unlabeled dataset $\mathcal{D}_s = \{x_i\}_{i=1}^{N_s}$, where $x_i \in \mathcal{X}$ for each $i$. The training process is iterative. The server model is learned by repeating the *client update* and the *server update* for each communication round $t \in \{1, 2, ..., T\}$.

**Client update** At round $t$, each client $c$ receives the server model $f_s^{t-1}$ from the previous round $t-1$ and initializes its client model with it. Then, each client $c$ trains its client model in a supervised way with each client's own labeled dataset $\mathcal{D}_c$; *e.g.*, with the cross-entropy loss $l$, each client $c$ trains its model by minimizing the empirical loss $f_c^t = \arg \min_f \mathcal{L}_c(f)$ for $E_c$ epochs, where $\mathcal{L}_c(f) := \mathbb{E}_{(x,y) \sim \mathcal{D}_c}[l(f(x), y)]$. Then, all clients transfer their models $\{f_c^t\}_{c=1}^C$ to the server.

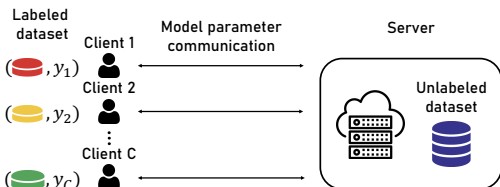

Figure 2: Problem setting. Each client has a labeled dataset, whereas a central server has an unlabeled dataset. The server model is learned through iterative communications of model parameters between the clients and the server.

**Server update** In the server update phase, at round $t$, the server aggregates the client models $\{f_c^t\}_{c=1}^C$ into a single server model $f_s^t$. Distinctively, we aggregate the client models by leveraging unlabeled server dataset $\mathcal{D}_s$. Designing such a model aggregation method is our goal of this work. After we establish the server model $f_s^t$, the model is distributed to all the clients.

### 3.2 FEDERATED LEARNING VIA UNCERTAINTY-AWARE ENSEMBLE DISTILLATION AND SELF-SUPERVISION

We focus on the client model aggregation step in the server update phase. In the server update phase, the server model starts either from the previous model or from the parameter average of client models. Then, our FedDS (Alg. 1) aggregates client models' knowledge via an ensemble distillation that aggregates client models while taking into account uncertainty of each client model per sample as a prediction quality, called entropy-weighted ensemble distillation (EED). We also propose to exploit unlabeled server dataset by self-supervised learning (SSL), so that the server model can further learn generic knowledge about the task. Both our proposed methods exploit unlabeled dataset. This stabilizes and better poses the server model as better initial points to clients at every round. We simultaneously conducts EED and SSL with the unlabeled server data.

**Entropy-weighted ensemble distillation (EED)** A pseudo label $l_p$ for each unlabeled data $x$ is generated by normalizing the entropy-weighted sum of predictions of each client model as:

$$l_p = \frac{\sum_{c=1}^C f_c^t(x) \exp\left(-k H\left(f_c^t(x)\right)\right)}{\left\|\sum_{c=1}^C f_c^t(x) \exp\left(-k H(f_c^t(x))\right)\right\|_1},$$

---

**Algorithm 1: FedDS on $C$ clients for $T$ rounds.** Experimentally, when $N_c \leq N_s$, setting $B_{AVG} = false$ gives higher test accuracy (refer Table 2 and Table 3). $f \circ g$ denotes the composition of functions $f$ and $g$.

⟨Notations⟩

$\mathcal{D}_s$: an unlabeled dataset in the server.   $\mathcal{D}_c$: a labeled dataset in the $c$-th client.

$f_s^t = \{f_{s_f}^t, f_{s_c}^t, f_{s_s}^t\}$: the server model at round $t$, where $f_{s_f}^t$ denotes a feature extracter part of the server model, $f_{s_c}^t$ a classifier part of the server model, and $f_{s_s}^t$ a self-supervised learning (SSL) classifier of the server model, which is only used to calculate the SSL loss.

$f_c^t$: the $c$-th client model at round $t$.

$E_s$: the number of server training epochs for each round.   $\gamma$: a weight hyperparameter for the SSL loss.

$B_{AVG}$: a boolean hyperparameter that determines the server's initial model for each round.

---

**Input** : $\mathcal{D}_s, \mathcal{D}_c$ for $c = 1, \dots C$

Randomly initialize $f_1^0, \dots f_C^0, f_s^0$

**for** *round $t = 1, \dots T$* **do**

    **for** *client $c = 1 \dots C$* **in parallel do**

        $f_c^t \leftarrow$ ClientUpdate $(f_{s_c}^{t-1} \circ f_{s_f}^{t-1}, \mathcal{D}_c)$

    **end**

    **if** $B_{AVG} = true$ **then**

        $f_s^t \leftarrow \frac{1}{C} \sum_{c=1}^{C} f_c^t$              `/* start from average of client models */`

    **else**

        $f_s^t \leftarrow f_s^{t-1}$                   `/* start from previous server model */`

    **end**

    **for** *epoch $e = 1, \dots E_s$* **do**

        **for** *each input data $x \in \mathcal{D}_s$* **do**

            $l_p \leftarrow$ CreatePseudoLabel $(f_1^t, \dots, f_C^t, x)$

            $\mathcal{L}_{CE} \leftarrow$ CrossEntropy $(l_p, (f_{s_c}^t \circ f_{s_f}^t)(x))$

            $\mathcal{L}_{SSL} \leftarrow$ GetSelfSupervisedLoss $(f_{s_s}^t \circ f_{s_f}^t, x)$

            Optimize $f_s^t$ to minimize $\mathcal{L}_{CE} + \gamma \mathcal{L}_{SSL}$         `/* using any optimizer */`

        **end**

    **end**

**end**

**return** $f_{s_c}^T \circ f_{s_f}^T$

---

where $H(p) = -\sum_i p_i \log p_i$ for a probability vector $p$ denotes the entropy of $p$, $k$ the temperature hyperparameter, and $\|\cdot\|_1$ the $L_1$-norm. The entropy $H(f_c^t(x))$ measures the degree of uncertainty about the class prediction $f_c^t(x)$ made by client $c$ for input $x$ at round $t$: the larger the entropy is, the larger the uncertainty is. Then, the pseudo label is used to compute the cross-entropy loss $\mathcal{L}_{CE}$ with the server model's prediction $(f_{s_c}^t \circ f_{s_f}^t)(x)$ as done in the standard knowledge distillation (Hinton et al., 2015).

The entropy-based weighting mitigates the influence of unreliable clients by suppressing the uncertain predictions. Our preliminary experiment in Fig. 3 shows that this entropy-based weighting indeed tends to assign higher weights to more correct predictions, i.e., the predictions for which softmax values on the true classes are higher. In Fig. 3, we compared with another weighting method based on maximum softmax values of clients's prediction, which does not exhibit such a tendency.

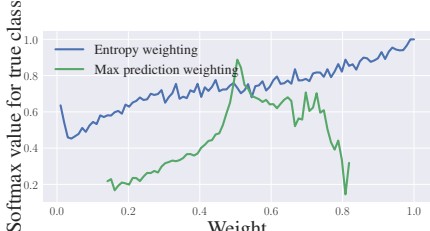

Figure 3: In CIFAR-10 classification task, the relationship between the assigned weight and the initial softmax value on the true class for each client is illustrated. The blue curve is based on our entropy-based weighting method, and the green curve is based on the max prediction weighting method using the maximum softmax value for comparison. The weight $\exp\left[-kH\left(f_c^t(x)\right)\right]$ with $k=5$ for entropy weighting and $\max_{class} f_c^t(x)$ for max prediction weighting are normalized so that the weights for all the clients for given unlabeled data sum to 1. The range of weight $[0, 1]$ is quantized into 50 bins and the weight of each client for each data $x$ is mapped to the nearest bin. For each bin, the softmax values on the true classes are averaged.

Table 1: Comparison between FedDS and other baseline algorithms.

|  | FedAVG | FedDF | FedD (ours) | FedAVG-S | FedDS (ours) |
|---|---|---|---|---|---|
| Ensemble distillation |  | ✓ | ✓ |  | ✓ |
| Uncertainty awareness |  |  | ✓ |  | ✓ |
| Self-supervised loss |  |  |  | ✓ | ✓ |

**Self-supervised learning**  The EED is applied jointly with self-supervised learning (SSL) to further leverage the benefits of available unlabeled data. An arbitrary SSL method can be applied, which helps to learn generic feature representation (Chen & He, 2021; Chen et al., 2020; Gidaris et al., 2018). This may act as stabilizing the FL learning procedure at aggregation, which motivates us to deploy. For image data, we use the image rotation prediction (Gidaris et al., 2018) as the self-supervised pretext task, *i.e.*, rotating input images and predicting those rotated angles, which is known to be simple and effective without cumbersome tuning. We use it due to its simplicity, but SSL in our approach is not limited to that method. We append the SSL head $f_{s_s}^t$ on top of the server's feature extractor part $f_{s_f}^t$, *i.e.*, $(f_{s_s}^t \circ f_{s_f}^t)(x)$. The SSL loss is applied to the output of the SSL head $f_{s_s}^t$. Our final loss for aggregating client models into the server model is defined as $\mathcal{L}_{CE} + \gamma \mathcal{L}_{SSL}$ in a multi-task manner.

## 4 EXPERIMENT RESULTS

### 4.1 EXPERIMENT SETUP

**Dataset and model**  Our FedDS is evaluated on CIFAR-10/100 and PathMNIST datasets with the ResNet-18 (He et al., 2016). CIFAR-10 (resp. CIFAR-100) has 10 classes (resp. 100 classes) containing 6,000 images (resp. 600 images) per class. For each of CIFAR-10/100 datasets, the training and test sets contain 50,000 and 10,000 images, respectively. PathMNIST (Yang et al., 2021) is a medical image dataset consisting of 9 classes of Colon Pathology images. The training and test sets contain 89,996 and 7,180 images, respectively.

**Dataset split**  In our experimental setting, we mainly assume 4 clients unless otherwise mentioned. The total client training dataset is sampled in such a way that it has the same class distribution as the entire training dataset, and it is partitioned by the number of clients and allocated to each client. Each client's class distribution follows the Dirichlet distribution Dir($\alpha$) as in Lin et al. (2020; 2021); Yurochkin et al. (2019); Wang et al. (2020); Hsu et al. (2019). The hyperparameter $\alpha$ indicates the homogeneity of the distribution: the higher $\alpha$ is, the closer to i.i.d. client data setting. We use $\alpha = 100,000$ and $\alpha = 0.5$ for the i.i.d. and non-i.i.d. client data settings, respectively. Some portion of a training dataset not sampled for client training datasets is used in the server without labels to mimic an unlabeled dataset.

**Baseline algorithms**  Our baseline algorithms are FedProx, FedDF (Lin et al., 2020), and variants of FedAVG and our FedDS. Since FedAVG (McMahan et al., 2017) and does not leverage unlabeled data, we devise a method by combining FedAVG and SSL (called FedAVG-S). After the server model is updated by model averaging in FedAVG, we apply the SSL loss (the rotation loss (Gidaris et al., 2018)) to further update the server model with the unlabeled dataset on the server. We confirm that this indeed provides a strong baseline in the presence of unlabeled data on the server (refer to Sec. 4.2). In addition, to see the effect of the SSL task in our FedDS, we consider a variant of our FedDS (called FedD), where the SSL part is suppressed by setting the SSL loss weight $\gamma = 0$. For FedD and FedDS, we set $B_{AVG} = false$ unless otherwise mentioned.

The main characteristics of the baselines and FedDS are summarized in Table 1. In summary, both FedDF and FedD are based on ensemble distillation, where FedD incorporates entropy-based weights to take into account confidence of each client. On the other hand, FedAVG-S and FedDS both apply SSL, while FedDS takes the ensemble distillation jointly with SSL.

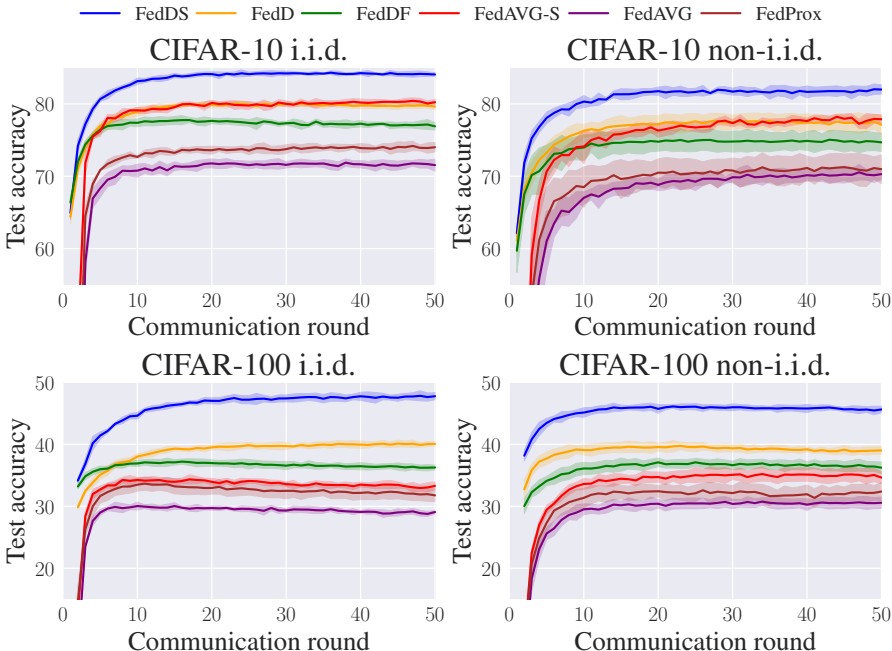

Figure 4: Test accuracy of FedDS and baseline algorithms according to communication rounds on CIFAR-10/100 classification tasks with i.i.d. and non-i.i.d. client data settings. Each client has 2,500 labeled data and the server has 40,000 unlabeled data. Each experiment was repeated 5 times by changing random seeds. Each line and shaded region represent the mean and standard deviation of the repeated experiments, respectively.

## 4.2 RESULTS WITH NORMAL DATA CONDITION

In this subsection, we present the accuracy results for CIFAR-10/100 and PathMNIST classification tasks with i.i.d. and non-i.i.d. client data settings, called normal data condition. For CIFAR-10/100 tasks, Fig. 4 shows the test accuracy of FedDS and baselines according to the communication rounds. It shows that the proposed FedDS consistently outperforms all the baselines starting from the early communication rounds. The effectiveness of our algorithm according to the number of clients and the size of client data is analyzed based on Tables 2 and 3. Table 4 shows that our FedDS still outperforms the baselines even when unlabeled data on the server is relatively small. Finally, the effectiveness of our algorithm on motivating medical diagnosis applications is verified via the PathMNIST classification task in Table 5.

Table 2: Test accuracy of FedDS and baselines at the 50th communication round on CIFAR-10 classification task for various numbers of clients with the same sized and i.i.d. client data setting. Each client has 1,000 labeled data and the server has 10,000 unlabeled data. For $B_{AVG} = false$, the performance gain due to EED is evaluated by the difference between the test accuracy of FedD and that of FedAVG, the additional gain due to SSL is calculated by the difference between the test accuracy of FedDS and that of FedD. The performance difference of FedDS by taking client model average is calculated by the difference between the test accuracy of FedDS with $B_{AVG} = true$ and that with $B_{AVG} = false$.

| Clients | Total number of client data | FedAVG | FedD (ours) | FedDS (ours) | |
|---------|------------------|--------|-----------------|-----------------|-----------------|
| | | | $B_{AVG} = false$ | $B_{AVG} = false$ | $B_{AVG} = true$ |
| 2 | 2,000 | 50.42 | 60.26 (+9.84) | **66.28** (+6.02) | 62.01 (-4.27) |
| 4 | 4,000 | 59.43 | 67.36 (+7.93) | **72.02** (+4.66) | 69.40 (-2.62) |
| 8 | 8,000 | 66.59 | 72.78 (+6.19) | **75.43** (+2.65) | 75.09 (-0.34) |
| 20 | 20,000 | 76.53 | 76.15 (-0.38) | 78.11 (+1.96) | **81.12** (+3.01) |

Table 3: Test accuracy of FedDS and baselines at the 50th communication round on CIFAR-10 classification task for fixed number of clients and various sizes of client datasets with i.i.d. client data setting. The server has 10,000 unlabeled data. The clients are trained with the same number of gradient step updates. For $B_{AVG} = false$, the performance gain due to EED is evaluated by the difference between the test accuracy of FedD and that of FedAVG, and the additional gain due to SSL is calculated by the difference between the test accuracy of FedDS and that of FedD. The performance difference of FedDS by taking client model average is calculated by the difference between the test accuracy of FedDS with $B_{AVG} = true$ and that with $B_{AVG} = false$.

| Clients | Total number of client data | FedAVG | FedD (ours) | FedDS (ours) | |
|---|---|---|---|---|---|
| | | | $B_{AVG} = false$ | $B_{AVG} = false$ | $B_{AVG} = true$ |
| 4 | 2,000 | 50.63 | 57.43 (+6.80) | **66.42** (+8.99) | 61.38(-5.04) |
| 4 | 4,000 | 59.43 | 67.36 (+7.93) | **72.02** (+4.66) | 69.40(-2.62) |
| 4 | 10,000 | 74.46 | 76.82 (+2.36) | **78.78** (+1.96) | 78.32(-0.46) |

**Comparison of two ensemble-distillation methods** We confirm the effectiveness of our uncertainty aware ensemble-distillation by comparing two ensemble-distillation methods, *i.e.*, our FedD and FedDF (Lin et al., 2020). For all the settings in Fig. 4, our FedD outperforms FedDF. The main differences of our FedD from FedDF are twofold; measuring the confidence of each client's prediction based on the entropy of softmax values and choosing the initial point of the ensemble distillation as the previous server model. For the initial point of the ensemble distillation, our FedD with $B_{AVG} = false$ starts ensemble distillation directly from the server model at the previous round while FedDF starts ensemble distillation from the parameter average of transferred client models at each round. Our empirical study in Tables 2 and 3 shows that using the averaged client model as a new initialization at each round is detrimental when the total number of client data is small.

**Effect of EED and SSL** In Fig. 4, our FedD shows comparable or better performance to the other baselines for all the settings, indicating that the entropy-weighted ensemble distillation is as effective as applying self-supervision, *i.e.*, FedAVG-S, or even better. From FedAVG-S results, it is interesting to see that applying SSL is mostly effective, but its effect is diminishing in some cases. FedAVG-S achieves the highest accuracy among the baselines in the CIFAR-10 classification task. However, it achieves similar accuracy with FedProx in the CIFAR-100 classification task. We conjecture that this is because the method of averaging the client model parameters in FedAVG results in unstable initialization for SSL when the total number of client data is small.

By jointly applying SSL with FedD, corresponding to our FedDS, we can see considerable performance improvements. In particular, our FedDS is effective for the CIFAR-100 classification task, where the data per each client is quite small, *i.e.*, 25 images per class on average.

Table 4: Test accuracy of FedDS and baselines at the 50th communication round on CIFAR-10 classification task with various sizes of unlabeled server dataset with i.i.d. client data setting. Each of four clients has 2,500 labeled data. The numbers of unlabeled server data for each algorithm are represented in the second row of the table.

| | FedAVG | FedDF | FedDS | | | |
|---|---|---|---|---|---|---|
| No. unlabeled server data | 0 | 40,000 | 10,000 | 20,000 | 30,000 | 40,000 |
| Test accuracy | 72.87 | 77.37 | 80.22 | 82.46 | 83.48 | 84.15 |

**Experiments with various numbers of clients and sizes of client data** To examine the effectiveness of FedDS with various dataset-split setups, we evaluate the performance gain of our FedDS compared to the baselines for various numbers of clients (Table 2) and for various sizes of dataset in each client (Table 3). For our FedDS with $B_{AVG} = false$, Table 2 shows that both the gain due to EED (in blue color) and the additional gain from SSL (in red color) increase as the number of clients decreases. For small number of clients, the influence of unreliable clients becomes higher. Thus, the EED which mitigates the influence of unreliable clients, and the SSL which provides the generic

Table 5: Test accuracy of FedDS and baseline algorithms measured at the 50th communication round on PathMNIST classification task with i.i.d. client data setting. Each client has 2,500 labeled data and the server has 40,000 unlabeled data. Each experiment was repeated 5 times by changing the random seed and the mean values are taken.

| Client data distribution | FedAVG | FedDF | FedAVG-S | FedDS (ours) |
|---|---|---|---|---|
| i.i.d. | 87.59 | 88.34 | 89.66 | **90.51** |
| non-i.i.d. | 86.80 | 87.06 | 88.93 | **90.05** |

knowledge about the task become more effective. Similarly, Table 3 shows that our FedDS with $B_{AVG} = false$ is more effective as the clients' data becomes more deficient. Particularly, we can see that SSL becomes the dominant factor of performance gain as the client's dataset gets smaller.

Tables 2 and 3 also show the performance difference of FedDS by setting $B_{AVG} = true$ instead of $B_{AVG} = false$ (represented in teal colors). It shows that starting from the server model at the previous round ($B_{AVG} = false$) performs better when the total number of client data is relatively small. This can be caused by the fact that averaging client models with limited client data unstabilizes the initial point, because a small fraction of outlier or undesirable contamination in client parameters can lead to notable biases in resultant parameter aggregation.

**Small unlabeled dataset at the server is still useful**  To see the effect of the size of the unlabeled server dataset on the performance gain of FedDS, FedDS was evaluated for the CIFAR-10 classification task by changing the size of server's unlabeled dataset. Table 4 shows that FedDS is still useful when the number of unlabeled data at the server is comparable with that of total labeled data at the clients. Although the test accuracy of FedDS decreases as the number of unlabeled data at the server decreases, FedDS using 10,000 unlabeled data still outperforms FedDF using 40,000 unlabeled data.

**Results on medical data**  To exemplify a medical diagnosis scenario with client-data deficiency, we evaluate the performance of the proposed FedDS and other baselines for PathMNIST classification task with i.i.d and non-i.i.d. client data settings. Table 5 confirm the effectiveness of our method.

### 4.3 RESULTS WITH UNRELIABLE CLIENT-SIDE DATA CONDITION

Table 6: Test accuracy of FedDS and baseline algorithms measured at the 50th communication round on CIFAR-10 classification task under various scenarios with unreliable clients. Each of four clients has 2,500 labeled data and the server has 40,000 unlabeled data. Noise-1 (resp. Noise-20) represents a noisy-label scenario where 1% (resp. 20%) of each client's labels is mislabeled. Byzantine represents a Byzantine attack scenario with a malicious client whose labels are totally mislabeled. The performance degradation in i.i.d. setting and in non-i.i.d. setting is evaluated by the difference between the test accuracy under normal data condition (No noise) and that under the corresponding noise scenario. Each experiment was repeated 5 times by changing the random seed and the mean values are taken.

| Noise type | Client data distribution | FL without SSL | | | FL with SSL | |
|---|---|---|---|---|---|---|
| | | FedAVG | FedDF | FedD (ours) | FedAVG-S | FedDS (ours) |
| No noise | i.i.d. | 71.53 | 76.90 | 79.64 | 80.24 | **84.07** |
| | non-i.i.d. | 70.29 | 74.69 | 77.12 | 77.88 | **81.99** |
| Noise-1 | i.i.d. | 65.54 ( -5.99) | 72.04 ( -4.86) | 75.05 ( -4.59) | 75.24 ( -5.00) | **81.14** ( -2.93) |
| | non-i.i.d. | 63.74 ( -6.55) | 69.32 ( -5.37) | 71.85 ( -5.27) | 72.73 ( -5.15) | **78.14** ( -3.85) |
| Noise-20 | i.i.d. | 50.78 (-20.75) | 55.31 (-21.59) | 59.88 (-19.76) | 59.06 (-21.18) | **67.94** (-16.13) |
| | non-i.i.d. | 45.04 (-25.25) | 47.95 (-26.74) | 52.37 (-24.75) | 55.11 (-22.77) | **60.30** (-21.69) |
| Byzantine | i.i.d. | 56.74 (-14.79) | 65.92 (-10.98) | 75.27 ( -4.37) | 66.12 (-14.12) | **80.30** ( -3.77) |
| | non-i.i.d. | 56.32 (-13.97) | 58.88 (-15.81) | 70.20 ( -6.92) | 64.16 (-13.72) | **80.27** ( -1.72) |

To see the robustness of uncertainty-aware ensemble distillation and self-supervision, we consider two types of scenarios with unreliable clients. In particular, we consider two noisy-label scenarios with 1%

and 20% label noises at each client (denoted as Noise-1 and Noise-20, respectively), and a Byzantine attack scenario with a malicious client whose labels are totally mislabeled (denoted as Byzantine). We present the test accuracy of FedDS and baselines for Noise-1, Noise-20, and Byzantine scenarios in Table 6. Let us focus on the i.i.d. client data setting as the tendency for the non-i.i.d. case is almost the same. While our FedD's performance degradation for Noise-1 and Noise-20 scenarios is slightly smaller than FedDF, it is remarkably smaller for the Byzantine case. This shows that the server model effectively rejects the Byzantine client's prediction using our entropy-based weighting (refer to Fig. 3). Our FedDS shows the most robust performance, *i.e.* performance degradations for Noise-1, Noise-20, and Byzantine senarios are the much smaller than other baselines.

## 5 CONCLUSION

In this work, we propose FedDS, which exploits the unlabeled data on the server side by entropy-weighted ensemble distillation (EED) and self-supervised learning (SSL). By using both techniques jointly, we demonstrate that FedDS outperforms the competing algorithms when the number of clients is small and data in the clients is deficient. In particular, when clients are unreliable, both techniques effectively suppress unreliable clients and reduce accuracy degradation. It is more effective as the number of participating clients are smaller, because, in such regimes, even introducing a single unreliable client would have much higher impact than other cases. Also, SSL effectively imposes the common knowledge for generic tasks to the server model. Thus, it stabilizes the overall learning round and helps the server model to overcome the deficiency of client data, especially when mislabeled samples are prevalent across clients. Our experiments show that EED and SSL lead to their own improvement and jointly using them still preserves notable improvement, which may imply that they play different roles that are both necessary for FL.

**Limitations**    The main limitation is that FedDS requires additional server resources including computation and additional unlabeled data. FedDS would be more fruitful on the situation when the central server has large unlabeled dataset and sufficient computation power.

**Further directions**    To apply FedDS to the situation when there is little unlabeled data on the server, some synthetic data would be useful. Lin et al. (2020) shows that the ensemble distillation using synthetic data from GAN gives decent pseudo label.

FedDS would also be applicable to the clients with heterogeneous architectures. For each round, one can update clients regardless of heterogeneous architectures by distilling the server model's knowledge using the client's data, instead of directly substituting the server model's parameter to each client model. FedDS would be applicable to tasks other than computer vision, like natural language. One can choose a suitable SSL task, like next word prediction, and use it as a SSL loss for FedDS.

**Reproducibility**    Our algorithms are described in detail at Sec. 3.2 and Sec. A.1.1. Implementation details needed to reproduce our results are described in Sec. 4.1 including hyperparameters and data distribution. Our source code will be publicized upon acceptance.

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

# A APPENDIX

The appendix consists of a detailed description of the proposed FedDS algorithm (Sec. A.1.1), additional experimental results (Sec. A.2), and dataset and model information (Sec. A.4).

## A.1 ALGORITHMS AND IMPLEMENTATION DETAILS

### A.1.1 ALGORITHM DESCRIPTION

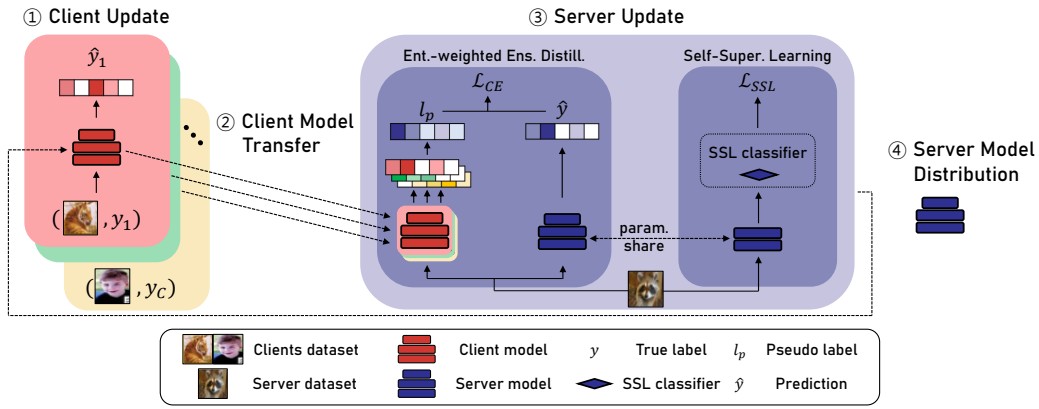

Figure 5: Illustration of the proposed FedDS algorithm. In each round, the server model is updated by simultaneously conducting entropy-weighted ensemble distillation (EED) and self-supervised learning (SSL) with the unlabeled server data.

Fig. 5 illustrates the operation of the proposed FedDS algorithm. Alg. 2, Alg. 3, and Alg. 4 correspond to ClientUpdate, CreatPseudoLabel, GetSelfSupervisedLoss algorithms used in the FedDS algorithm (Alg. 1), respectively.

Alg. 2 updates the client model as in FedAVG (McMahan et al., 2017). It trains the distributed server model using the client's own data and returns the trained model.

Alg. 3 creates the pseudo label for entropy-weighted ensemble distillation (EED). It gets the client models from Alg. 2 and creates a pseudo label for each unlabeled data of the server as the entropy-weighted average of the client models' predictions.

Alg. 4 illustrates an example of self-supervised learning (SSL) algorithm, which we used in the experiments. In general, GetSelfSupervisedLoss algorithm returns an SSL loss $\mathcal{L}_{SSL}$ for each unlabeled data $x$ of the server from feature extractor $f_{s_f}^t$ and SSL classifier $f_{s_s}^t$ of the server model. In this example, we assume that the input data is an image, and use the predicting image rotation task (Gidaris et al., 2018) as the self-supervised pretext task, *i.e.* rotating the input image and predicting its rotated angle.

### A.1.2 IMPLEMENTATION DETAILS AND TRAINING

For each round, each client model is trained for 30 epochs using its own dataset unless otherwise mentioned. We use the Adam optimizer with the client settings of learning rate 0.001, $\beta = (0.9, 0.999)$, weight decay 0, and batch size 64 for CIFAR-10/100 and 128 for PathMNIST[1]. In the server update phase in each round, the server model is trained for 15 epochs for CIFAR-10/100 and one epoch for PathMNIST using its own unlabeled dataset. The unlabeled dataset is pseudo-labeled by the gathered client models in the proposed entropy-weighted ensemble way. We set the entropy weight temperature $k = 5$ obtained from our empirical tests. We also use the Adam optimizer with the server settings of learning rate 0.00038 for CIFAR-10/100 and 0.001 for PathMNIST, $\beta = (0.9, 0.999)$, weight decay 0, and batch size 64 for CIFAR-10/100 and 128 for PathMNIST. The learning rate for

---

[1]The hyperparameters for PathMNIST in our experiments are chosen by following those of Yang et al. (2021).

---

**Algorithm 2:** ClientUpdate (same as the FedAVG).

$\mathcal{D}_c$ is client $c$'s labeled dataset. $f^t_{s_f}$ and $f^t_{s_c}$ are the feature extractor and the classifier parts of the server model at round $t$, respectively. $f^t_c$ is the client $c$'s model at round $t$. $E_c$ is the number of client training epochs for each round.

---

**Input:** $f^{t-1}_{s_c} \circ f^{t-1}_{s_f}$, $\mathcal{D}_c$
$f^t_c \leftarrow f^{t-1}_{s_c} \circ f^{t-1}_{s_f}$
**for** $e = 1, \dots E_c$ **do**
    **for** *input data and its label* $(x, y) \in \mathcal{D}_c$ **do**
        $\mathcal{L}_{CE} \leftarrow \text{CrossEntropy}(y, f^t_c(x))$
        Optimize $f^t_s$ to minimize $\mathcal{L}_{CE}$               /* using any optimizer */
    **end**
**end**
**return** $f^t_c$

---

---

**Algorithm 3:** CreatePseudoLabel.

$f^t_c$ is the client $c$'s model at round $t$. $x$ is an unlabeled server data. $H(p) = -\sum_i p_i \log p_i$ for a probability vector $p$ denotes the entropy of $p$, $k$ is the temperature hyperparameter for entropy weight, and $|| \cdot ||_1$ denotes the $L_1$-norm.

---

**Input:** $f^t_1, \dots, f^t_C$, $x$

**return** $\dfrac{\sum_{c=1}^{C} f^t_c(x) e^{-kH\left(f^t_c(x)\right)}}{\left\Vert \sum_{c=1}^{C} f^t_c(x) e^{-kH(f^t_c(x))} \right\Vert_1}$

---

---

**Algorithm 4:** GetSelfSupervisedLoss (rotation loss (Gidaris et al., 2018))

An example of SSL algorithm, which is used for FedDS in all the experiments in this paper. $f^t_{s_f}$ and $f^t_{s_s}$ are the feature extractor and the SSL classifier parts of the server model at round $t$, respectively. In this example, the output of $f^t_{s_s}$ is the prediction of rotation angle. For an input image $x$, $\text{rot}_d(x)$ rotates $x$ by $d$ degrees.

---

**Input:** $f^t_{s_s} \circ f^t_{s_f}$, $x$
$\mathcal{L}_{SSL} \leftarrow 0$
**for** $i = 0, 1, 2, 3$ **do**
    $\mathcal{L}_{SSL} \leftarrow \mathcal{L}_{SSL} + \text{CrossEntropy}(i, f^t_{s_s} \circ f^t_{s_f}(\text{rot}_{90i}(x)))$
**end**
**return** $\mathcal{L}_{SSL}$

---

CIFAR-10/100 is found by a hyperparameter search tool, Optuna (Akiba et al., 2019). The balance parameter for the SSL loss is set to $\gamma = 300$. The underlying reason why such a large $\gamma = 300$ was applied is because the classification loss is implemented in a way that all the batch losses add up, instead of averaging. If we take into account such an accumulation effect, the equivalent balance parameter was in fact 4.68 = 300/64 (64 is the server's batch size). For the chosen $\gamma$, the proportion of the rotation loss is about twice that of the classification loss at the 50th round for the i.i.d. setting in Fig. 4

### A.2 ADDITIONAL EXPERIMENTAL RESULTS

### A.2.1 SELF-SUPERVISION IS MORE EFFECTIVE THAN A FEW LABELED DATA

To see the effect of a few labeled data on the server, we change 10% of server's dataset to be labeled data. The server model performs ensemble distillation with this labeled data by adding the supervised cross-entropy loss. For each batch of pseudo-labeled data, labeled data is appended into the batch and is trained together.

Table 7: Test accuracy of FedDS and baseline algorithms measured at the 50th communication round on CIFAR-10 classification task when all the server data is unlabeled and when 10% of server data is labeled and the remaining server data is unlabeled.

| Server labeled data | Client data distribution | FedDF | FedD (ours) | **FedDS (ours)** |
|---|---|---|---|---|
| No labeled | i.i.d. | 76.90 | 79.64 | **84.07** |
| | non-i.i.d. | 74.69 | 77.12 | **81.99** |
| 10% labeled | i.i.d. | 79.41 | 81.31 | **84.80** |
| | non-i.i.d. | 77.36 | 80.36 | **83.62** |

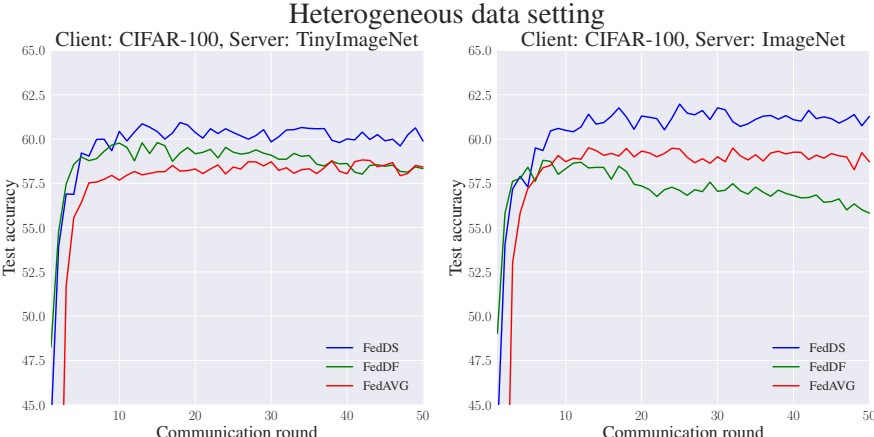

Figure 6: Test accuracy of FedDS, FedDF and FedAVG for CIFAR-100 classification task with 4 clients on i.i.d. client data settings. Each client has 12,500 labeled CIFAR-100 data and the server has 100,000 TinyImageNet unlabeled data (left) and 130,000 ImageNet (100 classes) unlabeled data (right).

Table 7 shows the effect of partial labels on the server. FedDS without labeled data outperforms both FedD and FedDF with labeled data. This shows that the self-supervision on large unlabeled data could give much valuable information than relatively little labeled data on our setting. More interestingly, the performance gain by using labels in FedDS is marginal. This may provide a hint at a possibility that the type of information from self-supervision is similar to and already covers the type of information the model can extract from labeled data.

### A.2.2 HETEROGENEOUS DATA SETTING

To see the effectiveness of our method under heterogeneous data settings, FedDS was evaluated with other baselines in Fig. 6 by using CIFAR-100 dataset for clients' labeled dataset and TinyImageNet (Le & Yang) and ImageNet(Krizhevsky et al., 2012) datasets for server's unlabeled dataset. In this experiment, the whole CIFAR-100 training dataset and the whole TinyImageNet/ImageNet training datasets were used for client training and for server training, respectively, and the test accuracy was evaluated for CIFAR-100 test dataset. Fig. 6 shows that FedDS outperforms both FedDF and FedAVG in such heterogeneous data settings.

### A.2.3 EXPERIMENT WITH A LARGE NUMBER OF CLIENTS

While our FedDS mainly targets FL scenarios where the number of clients or the labeled data of clients is insufficient, let us investigate the effectiveness of our method in a normal FL situation with a large number of clients. For CIFAR-10 classification task with 100 clients, Fig. 7 shows that FedDS and FedDF using ensemble distillation consistently outperform FedAVG from early communication rounds. The difference between FedDF and FedAVG shows that in situations where there are many clients, the effect of distillation decreases as the communication round increases. In contrast, the consistent gap between FedDS and FedDF shows that entropy-based weighting and SSL are effective even when there are many clients.

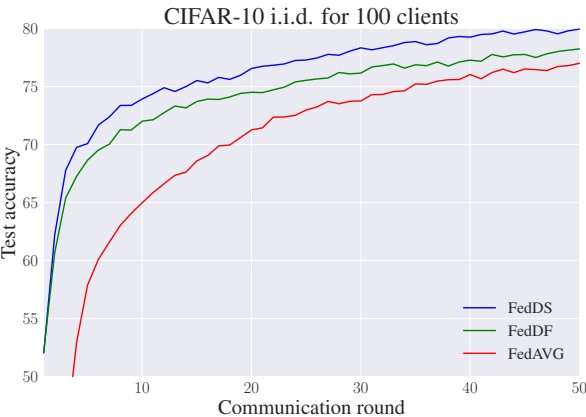

Figure 7: Test accuracy of FedDS, FedDF and FedAVG for CIFAR-10 classification task with 100 clients on i.i.d. client data settings. Each client has 400 labeled data and the server has 10,000 unlabeled data.

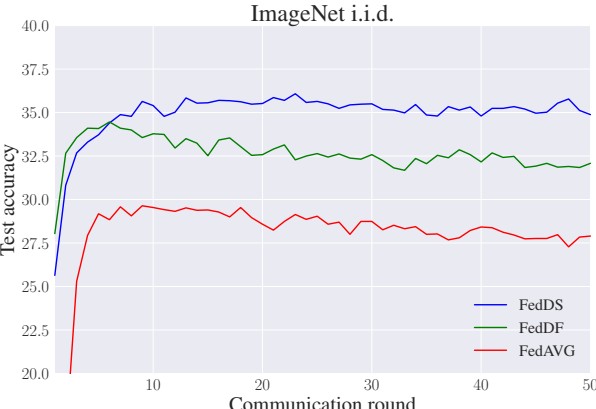

Figure 8: Test accuracy of FedDS, FedDF and FedAVG for ImageNet classification task with 4 clients on i.i.d. client data settings. Each client has 6,500 labeled data and the server has 104,000 unlabeled data.

### A.2.4 EXPERIMENT WITH A LARGER DATASET

Fig. 8 shows that our FedDS outperforms FedDF and FedAVG for ImageNet classification task (Krizhevsky et al., 2012), which has 100 classes, 130,000 images for training and 5,000 images for test. For computational feasibility, we down-sampled to image resolution 32 (Chrabaszcz et al., 2017).

### A.2.5 EXPERIMENT WITH DIFFERENT CLASS DISTRIBUTIONS AT THE CLIENTS AND AT THE SERVER

In our main experiments in Sec. 4, we assumed that the class distribution of the whole client dataset and that of the server's unlabeled dataset are the same. To investigate the effectiveness of our method when those class distributions are different, we evaluated FedDS with other baselines in Fig. 9 for CIFAR-10 classification task with i.i.d. class distribution for clients and Dir(0.5) distribution for the server. We can see that our FedDS outperforms the baselines in such scenarios with different class distributions at the clients and at the server.

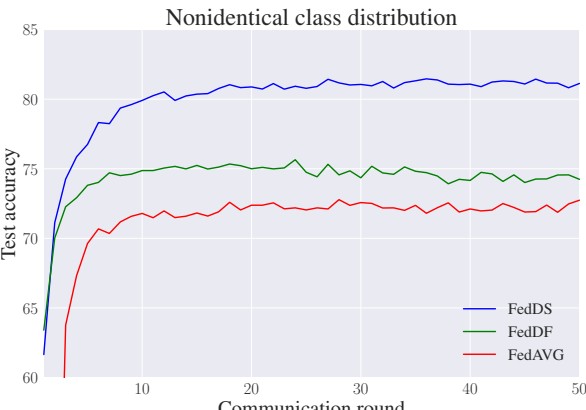

Figure 9: Test accuracy of FedDS, FedDF and FedAVG for CIFAR-10 classification task with 4 clients on i.i.d. client data settings. Each client has 2,500 labeled data and the server has 20,000 unlabeled data. Server data follows Dirichlet($\alpha$) distribution for $\alpha = 0.5$.

Table 8: Test accuracy of FedDS at the 50th communication round on CIFAR-10 classification task with different methods of combining EED and SSL. In addition to the proposed method of simultaneously conducting EED and SSL (2nd row), a sequential approach is considered where SSL starts after EED (3rd row). In this experiment, the setting in Sec. 4.1 is assumed and the predicting image rotation task (Gidaris et al., 2018) is used as the SSL task.

| SSL timing | FedDS |
|---|---|
| Simultaneous | 84.07 |
| Sequential | 83.88 |

### A.2.6 ABLATION STUDIES

**Effect of combining methods for EED and SSL** In the proposed FedDS algorithm (Alg. 1), the server model is updated by applying the entropy-weighted ensemble distillation (EED) and the self-supervised learning (SSL) simultaneously. Another possible approach is to start SSL training after the end of EED training. Table 8 shows that the performance gap between the two methods of combining EED and SSL is marginal, indicating that the timing of SSL is not an important factor.

**Effect of temperature** Fig. 10 shows the optimized temperature values $k$ for FedDS from the hyperparameter search tool Optuna (Akiba et al., 2019). For the i.i.d. and the non-i.i.d. data settings, $k$ around 0.1 and $k$ around 8 achieve near-optimal performances, respectively, while the performance remains almost the same over a relatively wide range of $k$. In the non-i.i.d data setting, client's data deficiency is much severe than the i.i.d. case for certain classes, which necessitates more stronger effect of entropy-weighting that filters out immature predictions.

### A.3 THEORETICAL RESULT

We extend the generalization error bound presented in FedDF (Lin et al., 2020) from uniform averaging logits to a weighted-ensemble version corresponding to our FedD on Theorem 1. Our new analysis may hint the following important remarks: 1) the bound of FedDF (uniform averaging logits) is a special case of our bound, and 2) our entropy-weighted method (FedD) has the same upper bound as that of FedDF; thus, there is no reason that FedDF performs better than our FedD. The latter is also supported by our empirical experiments showing that our FedD outperforms FedDF, e.g., in Fig. 4 and Table 6.

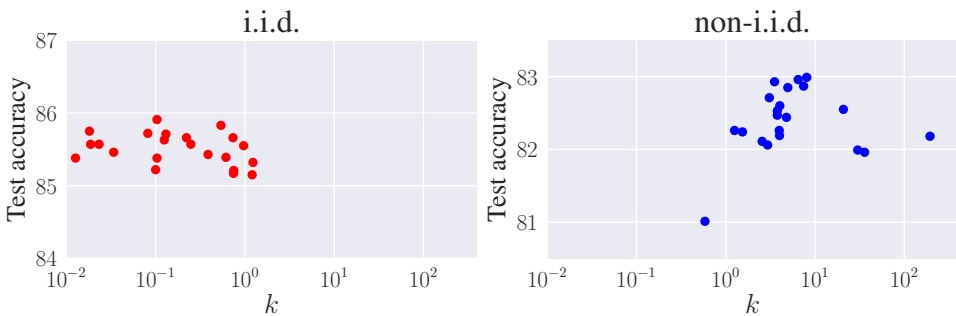

Figure 10: Test accuracy of FedDS on CIFAR-10 classification task for near-optimal values of entropy-weight temperature $k$ searched from the hyperparameter search tool Optuna (Akiba et al., 2019).

**Theorem 1** (informal) *We denote the global data distribution $\mathcal{D}$, the $c$-th local distribution $\mathcal{D}_c$, its empirical distribution $\hat{\mathcal{D}}_c$ with size $n$, and the global empirical distribution $\hat{\mathcal{D}} = \cup_c \hat{\mathcal{D}}_c$. The hypothesis $h \in \mathcal{H}$ learned on $\hat{\mathcal{D}}_c$ is denoted by $h_{\hat{\mathcal{D}}_c}$. Given the same assumptions with Lin et al. (2020), for every $h \in \mathcal{H}$ and any $\delta \in (0,1)$, with the probability $1 - \delta$, we have*

$$\mathbb{E}_{x \sim \mathcal{D}} \left[ \mathcal{L} \left( \sum_c w_c(x) \cdot h_{\hat{\mathcal{D}}_c}(x) \right) \right] \leq \mathbb{E}_{x \sim \hat{\mathcal{D}}} \left[ \mathcal{L} \left( h_{\hat{\mathcal{D}}}(x) \right) \right] + \sum_c m(\delta, n) + \frac{1}{2} d_{\mathcal{H} \Delta \mathcal{H}}(\mathcal{D}_c, \mathcal{D}) + \lambda_c$$

(1)

*where $w_c(\cdot)$ denotes our entropy weight for each clients with $\sum_c w_c(\cdot) = 1$ and $w_c(\cdot) \geq 0$, $d_{\mathcal{H} \Delta \mathcal{H}}(\cdot, \cdot)$ measures the distribution discrepancy between two distributions, $m(\delta, n) = \frac{4 + \sqrt{\log(\tau_{\mathcal{H}}(2n))}}{\delta \sqrt{2n}}$, $\lambda_c = \inf_h \mathbb{E}_{x \sim \mathcal{D}} [w_c(x) \mathcal{L}(h(x))] + \mathbb{E}_{x \sim \mathcal{D}_c} [w_c(x) \mathcal{L}(h(x))]$, and $\tau_{\mathcal{H}}$ is the growth function bounded by a polynomial of the VCdim of $\mathcal{H}$.*

*Proof sketch.* Given our entropy-weighted ensemble model $\sum_c w_c(x) \cdot h_{\hat{\mathcal{D}}_c}(x)$, we define its generalization error as $\mathbb{E}_{x \sim \mathcal{D}} \left[ \mathcal{L} \left( \sum_c w_c(x) \cdot h_{\hat{\mathcal{D}}_c}(x) \right) \right]$. Then, we have

$$\mathbb{E}_{x \sim \mathcal{D}} \left[ \mathcal{L} \left( \sum_c w_c(x) \cdot h_{\hat{\mathcal{D}}_c}(x) \right) \right] \leq \sum_c \mathbb{E}_{x \sim \mathcal{D}} \left[ w_c(x) \mathcal{L} \left( h_{\hat{\mathcal{D}}_c}(x) \right) \right] \quad \text{(By Jensen's inequality)}$$

$$= \sum_c \mathbb{E}_{x \sim \mathcal{D}} \left[ \mathcal{L}'_c \left( h_{\hat{\mathcal{D}}_c}(x) \right) \right] \quad (\mathcal{L}'_c(X) = w_c(x) \mathcal{L}(X))$$

$$\leq \sum_c \mathbb{E}_{x \sim \mathcal{D}_c} \left[ \mathcal{L}'_c \left( h_{\hat{\mathcal{D}}_c}(x) \right) \right] + \frac{1}{2} d_{\mathcal{H} \Delta \mathcal{H}}(\mathcal{D}_c, \mathcal{D}) + \lambda_c$$

(By the domain gap bound (Ben-David et al., 2010) with prob.)

$$\leq \sum_c \mathbb{E}_{x \sim \hat{\mathcal{D}}_c} \left[ \mathcal{L}'_c \left( h_{\hat{\mathcal{D}}_c}(x) \right) \right] + m(\delta, n) + \frac{1}{2} d_{\mathcal{H} \Delta \mathcal{H}}(\mathcal{D}_c, \mathcal{D}) + \lambda_c$$

(By the uniform convergence (Shalev-Shwartz & Ben-David, 2014) with prob.)

$$\leq \sum_c \mathbb{E}_{x \sim \hat{\mathcal{D}}_c} [w_c(x)^p]^{\frac{1}{p}} \mathbb{E}_{x \sim \hat{\mathcal{D}}_c} \left[ \mathcal{L} \left( h_{\hat{\mathcal{D}}_c}(x) \right)^q \right]^{\frac{1}{q}} + \sum_c H_c$$

(by Hölder's inequality and $H_c = m(\delta, n) + \frac{1}{2} d_{\mathcal{H} \Delta \mathcal{H}}(\mathcal{D}_c, \mathcal{D}) + \lambda_c$)

$$\leq \sum_c \mathbb{E}_{x \sim \hat{\mathcal{D}}_c} \left[ \mathcal{L} \left( h_{\hat{\mathcal{D}}_c}(x) \right) \right] + \sum_c H_c$$

(by $w_c(x) \in [0, 1]$ and setting $p = \infty$ and $q = 1$)

$$\leq \sum_c \mathbb{E}_{x \sim \hat{\mathcal{D}}_c} \left[ \mathcal{L} \left( h_{\hat{\mathcal{D}}}(x) \right) \right] + \sum_c H_c$$

(by the definition of ERM, $\mathcal{L}(h_{\hat{\mathcal{D}}_c}(x)) \leq \mathcal{L}(h_{\hat{\mathcal{D}}}(x))$)

$$= \mathbb{E}_{x \sim \hat{\mathcal{D}}} \left[ \mathcal{L} \left( h_{\hat{\mathcal{D}}}(x) \right) \right] + \sum_c H_c. \quad \text{(by } \hat{\mathcal{D}} = \cup_c \hat{\mathcal{D}}_c \text{)}$$

Thus, we have

$$\mathbb{E}_{x \sim \mathcal{D}}\left[\mathcal{L}\left(\sum_c w_c(x) \cdot h_{\hat{\mathcal{D}}_c}(x)\right)\right] \leq \mathbb{E}_{x \sim \hat{\mathcal{D}}}\left[\mathcal{L}\left(h_{\hat{\mathcal{D}}}(x)\right)\right] + \sum_c m(\delta, n) + \frac{1}{2}d_{\mathcal{H}\Delta\mathcal{H}}(\mathcal{D}_c, \mathcal{D}) + \lambda_c, \quad (2)$$

with a probability at least $1 - \delta$, which concludes the proof. □

## A.4 DATASET AND MODEL

For classification tasks, CIFAR-10/100 datasets (Krizhevsky, 2009)(MIT licence), ImageNet dataset (Krizhevsky et al., 2012) (100 classes), TinyImageNet (Le & Yang) dataset are used CIFAR-10 (resp. CIFAR-100) has 10 classes (resp. 100 classes) containing 6000 images (resp. 600 images) per class. ImageNet (100 classes) has 100 classes containing 1,300 (resp. 50) images per class for training (resp. test). TinyImageNet has 200 classes containing 500, 50, 50 images per class for training, validation, test, respectively. We used only validation data for evaluate the test accuracy. PathMNIST (Yang et al., 2021) is a medical image dataset consisting of 9 classes of Colon Pathology images. The training and test sets contain 89,996 and 7,180 images, respectively. The proposed FedDS and baseline algorithms are implemented on ResNet-18 (He et al., 2016) by changing the last layer to adjust to the number of classes, using the open source `https://github.com/kuangliu/pytorch-cifar.git`.

