# OpenReview forum: "Exploit Unlabeled Data on the Server! Federated Learning via Uncertainty-aware Ensemble Distillation and Self-Supervision"
_ICLR.cc/2023/Conference — Submitted to ICLR 2023_

### Official Review · Reviewer_CynN · 2022-10-23

**Confidence:** 4
**Correctness:** 3
**Technical Novelty And Significance:** 2
**Empirical Novelty And Significance:** 2
**Recommendation:** 3

**Clarity, Quality, Novelty And Reproducibility:**

- The paper is easy to follow, but some details are missing or unclear.
- The novelty of the paper is limited.
- The code is not provided.

**Strength And Weaknesses:**

Strengths:
1. The motivation is good. It is intuitive to suppress the contribution of unreliable clients.
2. The overall paper is well organized and easy to follow.

Weaknesses:
1. The novelty of the proposed method is limited. The overall method is FedDF + entropy-weighted + SSL loss. For ‘entropy-weighted’, the authors should explain more why entropy can be used to measure the uncertainty of classification. For ‘SSL loss’, in learning with unlabeled data, SSL loss is commonly used and can be added in any distillation-based model aggregation method. These two components added to FedDF are incremental.
2. The experiments are unconvincing. In most experimental settings, there are only four clients. Unless in the cross-silo setting, there should be at least 50 clients in the cross-device setting.
3. The proposed method needs to be tested on a larger dataset, the conclusion on CIFAR is not convincing.
4. The results shown in Figure 3 seems meaningless. The relationship between ‘softmax value for true class’ and ‘entropy-weight’ is inherent (because the former is calculated using the latter). How does this result relate to the motivation or performance of FedDS?
5. It is mentioned in 4.1 EXPERIMENT SETUP that ‘the total client training dataset is sampled in such a way that it has the same class distribution as the entire training dataset’, which means that the server dataset has the same distribution as those of all the clients. Under such a setting, it is not surprising that the scheme based on weighted distillation is effective, because the difference between the dataset used in distillation and the client datasets is too small.

**Summary Of The Paper:**

This paper proposes FedDS to improve the performance of the global model with distillation-based model aggregation. The key idea is to suppress the contributions of unreliable clients when performing multi-teacher ensemble distillation. Compared to FedDF, the authors added two additional components, the entropy-weighted pseudo label and the self-supervised learning (SSL) loss. The former calculates the uncertainty of each teacher's prediction score for weighting a teacher’s output. The latter can be an arbitrary SSL method. The experimental results verify the effectiveness of the FedDS to some extent.

**Summary Of The Review:**

The paper has a good motivation. However, the novelty (FedDF + entropy-weighted + SSL loss) is limited. The experiments are unconvincing that in most experiments there are only four clients and the dataset is small.

---

> ### Author Response · Authors · 2022-11-19
> **Response to reviewer CynN (2/2)**
>
> ### The experiments are unconvincing. In most experimental settings, there are only four clients. Unless in the cross-silo setting, there should be at least 50 clients in the cross-device setting.
>
> In accordance with the reviewer’s suggestion, we additionally investigated the effectiveness of our method in a normal FL situation with a large number of clients in Section A.2.3, while our FedDS mainly targets FL scenarios where the number of clients or the labeled data of clients is insufficient. For CIFAR-10 classification task with 100 clients, Fig. 7 shows that FedDS and FedDF using ensemble distillation consistently outperform FedAVG from early communication rounds. The difference between FedDF and FedAVG shows that in situations where there are many clients, the effect of distillation decreases as the communication round increases. In contrast, the consistent gap between FedDS and FedDF shows that entropy-based weighting and SSL are effective even when there are many clients.
>
> ### The proposed method needs to be tested on a larger dataset, the conclusion on CIFAR is not convincing.
>
> In accordance with the reviewer’s comment, we performed additional experiments for ImageNet classification task, which has 100 classes, 130,000 images for training and 5,000 images for test, and reported the result in Section A.2.4 of the revised paper. Fig. 8 shows that our FedDS outperforms other baseline algorithms for such a larger dataset.
>
> ### The results shown in Figure 3 seems meaningless. The relationship between ‘softmax value for true class’ and ‘entropy-weight’ is inherent (because the former is calculated using the latter). How does this result relate to the motivation or performance of FedDS?
>
> Let us first clarify how we plot the blue curve in Fig. 3 (there was only the blue curve in the originally submitted paper). Let’s consider client *c*. For an unlabeled data *x*, the weight of client *c* is determined based on the entropy of the softmax values of client *c* about data *x* (refer to Algorithm 3 in Section A.1.1 for exact formula.). Hence, the entropy-based weights are calculated based on the clients’ predictions (softmax values). Now, the blue curve of Fig. 3 is obtained based on the relationship between the entropy-based weight assigned to each client and the initial softmax value of that client on the true class for each unlabeled data.
> Through Fig. 3, we can justify that this entropy-based weighting indeed tends to assign higher weights to more correct predictions, i.e., the predictions for which softmax values on the true classes are higher. Although our entropy-based weighting is simple, it is not trivial to devise such a weighting method that assigns higher weights to more correct predictions. In Fig. 3, we compared with another weighting method (green curve) based on maximal softmax values of clients’s prediction, which does not exhibit such a tendency.
>
> ### It is mentioned in 4.1 EXPERIMENT SETUP that ‘the total client training dataset is sampled in such a way that it has the same class distribution as the entire training dataset’, which means that the server dataset has the same distribution as those of all the clients. Under such a setting, it is not surprising that the scheme based on weighted distillation is effective, because the difference between the dataset used in distillation and the client datasets is too small.
>
> In accordance with the reviewer’s comment, we reported the following additional experiments in Sections A.2.5 and A.2.2 of the revised paper.
>
> 1. To see the effectiveness of our method when class distributions at the clients and at the server are different, we evaluated FedDS with other baselines in Fig. 9 for CIFAR-10 classification task with i.i.d. class distribution for clients and Dir(0.5) distribution for the server. We can see that our FedDS outperforms the baselines in such scenarios.
> 2. To see the effectiveness of our method under heterogeneous data settings, FedDS was evaluated with other baselines in Fig. 6 by using CIFAR-100 dataset for clients’ labeled dataset and TinyImageNet and ImageNet datasets for server’s unlabeled dataset. In this experiment, the whole CIFAR-100 training dataset and the whole TinyImageNet or ImageNet training datasets were used for client training and for server training, respectively, and the test accuracy was evaluated for CIFAR-100 test dataset. Fig. 6 shows that FedDS outperforms both FedDF and FedAVG in such heterogeneous data settings.

---

> ### Author Response · Authors · 2022-11-19
> **Response to reviewer CynN (1/2)**
>
> ### The novelty of the proposed method is limited. The overall method is FedDF + entropy-weighted + SSL loss. For ‘entropy-weighted’, the authors should explain more why entropy can be used to measure the uncertainty of classification. For ‘SSL loss’, in learning with unlabeled data, SSL loss is commonly used and can be added in any distillation-based model aggregation method. These two components added to FedDF are incremental.
>
> The novelty of our work is on the development of an effective method that deals with unreliable clients in data deficient regimes by exploiting unlabeled data on the server, while the focus of DF is not on data deficient regimes. The distinctive features of our method are summarized in the following.
>
> - DF assumed and used validation set (small labeled dataset) on the server, while our method does not require such a validation set.
> - For ensemble distillation with unreliable clients, an effective rejection of uncertain predictions of clients is crucial. To tackle this issue, we developed a novel entropy-weighted ensemble distillation to measure and reflect the uncertainty of client predictions for each data point. In general, the entropy becomes higher when the corresponding distribution is closer to uniform. Since a client’s prediction becomes more uniform over the classes when it is less confident, our EED assigns a lower weight to the prediction with higher entropy. Fig. 3 of the revised paper shows that this entropy-based weighting indeed tends to assign higher weights to more correct predictions, i.e., the predictions for which softmax values on the true classes are higher.  Although our entropy-based weighting is simple, it is not trivial to devise such a weighting method that assigns higher weights to more correct predictions. In Fig. 3, we compared with another weighting method (green curve) based on maximal softmax values of clients’s prediction, which does not exhibit such a tendency.  On the other hand, there was no uncertainty-measuring step in the ensemble distillation of FedDF.
> - Our algorithm suggests a guideline about the starting point of the ensemble distillation based on the dataset sizes at the clients and server. In particular, our experimental results show that in data deficient regime, it’s better to start ensemble distillation directly from the server model at the previous round, instead of starting from the parameter average of transferred client model as done in FedDF.
> - Our method combines self-supervised learning with ensemble distillation to fully exploit the unlabeled data, while FedDF only utilizes the unlabeled dataset on the server for ensemble distillation.  Our experiments show that EED and SSL lead to their own improvement and jointly using them still preserves notable improvement, which implies that they play different roles that are both necessary for FL.
>
> As such, our main target scenario is different from FedDF and our FedDS was elaborately devised to reliably deal with data deficient regime.

---

### Official Review · Reviewer_qQ7S · 2022-10-25

**Confidence:** 3
**Correctness:** 3
**Technical Novelty And Significance:** 2
**Empirical Novelty And Significance:** 2
**Recommendation:** 5

**Clarity, Quality, Novelty And Reproducibility:**

Clarity and Quality:

Overall clear and well written paper.

Novelty:

I am a little concerned about novelty as stated above.

Reproducibility:

Seems to be reproducible but no appendix or code is provided.

**Strength And Weaknesses:**

Strength:

1: The paper is well motivated. It is true that in realistic FL tasks, we might be able to collect a lot of unlabeled data on central server. Thus, it is a very important question on how we could utilize such unlabeled data.

2: The proposed method is simple but seems to be effective in the presented experiments. It is interesting that it is shown that small unlabeled dataset will still be useful.

Weakness:

1: The proposed method might not be novel enough. As stated by authors in related work section, both distillation training and SSL method have already been applied to federated learning. Authors only propose additional entropy weighted ensembles for distillation, which also might not be new in ensemble learning.

2: In experiment sections, if I understand correctly, authors only use 4 clients in general (and upto 20 clients in some experiments). This might not be enough number of clients to be a meaningful federated learning experiment (which should have at least 100 clients).

3: Authors should also include many other FL baselines: (1) authors should include algorithms that focus on solving non-iid FL, such as FedProx and Scaffold, in order to show the effectiveness of using additional unlabeled data (2) author should also compare with many other FL+SSL algorithms (some of which are already mentioned by authors in related work section).

4: Lack of theoratical analysis of the proposed method. But I understand this is minor point for the emperical paper.



**Summary Of The Paper:**

In this paper, authors explore how to utilize unlabeled data on server to improve upon current Federated Learning methods. Authors propose (1) using entropy-weighted ensemble of client models to create pseudo labels for distillation training of global model (2) directly using SSL method with unlabeled data to help to train global model. Authors show compared with previous baselines such as FedAVG, FedDF and FedAVG-S, their proposed method shows significant increase in final accuracy. Also, authors show that their method demonstrates more robust performance with unreliable client-side data conditions. Finally, authors show small unlabeled datasets on server is still useful.

**Summary Of The Review:**

I recommend weak rejection mainly due to lack of overall novelty and some questions on the experiment settings.

---

> ### Author Response · Authors · 2022-11-19
> **Respose to reviewer qQ7S (1/2)**
>
> ### The proposed method might not be novel enough. As stated by authors in related work section, both distillation training and SSL method have already been applied to federated learning. Authors only propose additional entropy weighted ensembles for distillation, which also might not be new in ensemble learning.
>
> The novelty of our work is on the development of an effective method that deals with unreliable clients in data deficient regimes by exploiting unlabeled data on the server. Most FL settings assumed unlabeled dataset at the clients. For a similar setting to ours (with unlabeled dataset on the server), FedDF was proposed while the focus of DF is not on data deficient regimes.
>
> The distinctive features of our method are summarized in the following.
>
> - For ensemble distillation with unreliable clients, an effective rejection of uncertain predictions of clients is crucial. To tackle this issue, we developed a novel entropy-weighted ensemble distillation to measure and reflect the uncertainty of client predictions for each data point. In general, the entropy becomes higher when the corresponding distribution is closer to uniform. Since a client’s prediction becomes more uniform over the classes when it is less confident, our EED assigns a lower weight to the prediction with higher entropy. Fig. 3 of the revised paper shows that this entropy-based weighting indeed tends to assign higher weights to more correct predictions, i.e., the predictions for which softmax values on the true classes are higher.  Although our entropy-based weighting is simple, it is not trivial to devise such a weighting method that assigns higher weights to more correct predictions. In Fig. 3, we compared with another weighting method (green curve) based on maximal softmax values of clients’s prediction, which does not exhibit such a tendency.  On the other hand, there was no uncertainty-measuring step in the ensemble distillation of FedDF.
> - Our algorithm suggests a guideline about the starting point of the ensemble distillation based on the dataset sizes at the clients and server. In particular, our experimental results show that in data deficient regime, it’s better to start ensemble distillation directly from the server model at the previous round, instead of starting from the parameter average of transferred client model as done in FedDF.
> - Our method combines self-supervised learning with ensemble distillation to fully exploit the unlabeled data, while FedDF only utilizes the unlabeled dataset on the server for ensemble distillation.  Our experiments show that EED and SSL lead to their own improvement and jointly using them still preserves notable improvement, which implies that they play different roles that are both necessary for FL.
> - DF assumed and used validation set (small labeled dataset) on the server, while our method does not require such a validation set.
>
> ### In experiment sections, if I understand correctly, authors only use 4 clients in general (and upto 20 clients in some experiments). This might not be enough number of clients to be a meaningful federated learning experiment (which should have at least 100 clients).
>
> In accordance with the reviewer’s comment, we additionally investigated the effectiveness of our method in a normal FL situation with a large number of clients in Section A.2.3, while our FedDS mainly targets FL scenarios where the number of clients or the labeled data of clients is insufficient. For CIFAR-10 classification task with 100 clients, Fig. 7 shows that FedDS and FedDF using ensemble distillation consistently outperform FedAVG from early communication rounds. The difference between FedDF and FedAVG shows that in situations where there are many clients, the effect of distillation decreases as the communication round increases. In contrast, the consistent gap between FedDS and FedDF shows that entropy-based weighting and SSL are effective even when there are many clients.

---

> ### Author Response · Authors · 2022-11-19
> **Respose to reviewer qQ7S (2/2)**
>
> ### Authors should also include many other FL baselines: (1) authors should include algorithms that focus on solving non-iid FL, such as FedProx and Scaffold, in order to show the effectiveness of using additional unlabeled data (2) author should also compare with many other FL+SSL algorithms (some of which are already mentioned by authors in related work section).
>
> Thanks for the suggestion. In accordance with the reviewer’s suggestion, we evaluated FedProx for the same settings as in Fig. 4 and included its accuracy in the same figure in the revised paper.
>
> Since FedProx does not utilize unlabeled data on the server, its performance is between FedAVG, which also does not utilize unlabeled data, and FedDF for all settings of Fig. 4. We note that our algorithm has no restriction on the learning method at the clients. Thus, in situations where FedProx is more advantageous than FedAVG, our algorithm can be further improved by applying the learning method at the clients suggested in FedProx.
>
> For the second point, the FL + SSL algorithms mentioned in the paper are proposed for the FL scenarios where there is unlabeled data at the clients. Hence, those are not compatible with our setting assuming unlabeled data only at the server.
>
> ### Lack of theoratical analysis of the proposed method. But I understand this is minor point for the emperical paper
>
> We have added a new theoretical generalization error bound analysis (Theorem 1) of our FedD in Section A.3.
>
> We extend the generalization error bound presented in FedDF (Lin et al., 2020) from uniform averaging logits to a weighted-ensemble version corresponding to our FedD. Our new analysis may hint the following important remarks: 1) the bound of FedDF (uniform weight ensemble) is a special case of our bound, and 2) our entropy-weighted method (FedD) has the same upper bound as that of FedDF; thus, there is no reason that FedDF performs better than our FedD. The latter is also supported by our empirical experiments showing that our FedD outperforms FedDF.
>
> We believe this allows us to improve our understanding of our algorithm’s behavior (see Fig. 4 and Table 6).
>
> Also, we postulate that when the discrepancy between client and server datasets is large, the SSL loss helps to bridge the data distribution gap. This would be an interesting future direction to explore the theoretical relationship with SSL and the domain gap in our setting.

---

### Official Review · Reviewer_fQQY · 2022-11-01

**Confidence:** 3
**Correctness:** 3
**Technical Novelty And Significance:** 2
**Empirical Novelty And Significance:** 3
**Recommendation:** 6

**Clarity, Quality, Novelty And Reproducibility:**

The paper is organized in an easy-to-read way and clarifies the main results well with empirical experiments. Novelty is a bit lacking.

**Strength And Weaknesses:**

Pros:
1. The usage of entropy to denote prediction uncertainty of each client
2. Ablation comparison of different components of the framework (EED and SSL) is well done

Cons:
1. The entropy-induced uncertainty is at the client level, but could it be that a local model performs well on part of the unlabelled data only?  Can this pseudo label be calculated on a datapoint level?
2. The choice of $\gamma$ (balance parameter for SSL loss) is not well justified
3. The claim when $N_c \leq N_s$, settig $B_{\text{AVG}}$ = false would give higher test accuracy is not well justified: when $N_c =N_s$, how could $B_{\text{AVG}}$ affect test accuracy? Would be nice to show it empirically.
4. How the train-test split is done on each client? Do they have the same distribution? This needs to be detailed about.

**Summary Of The Paper:**

This paper proposed a knowledge distillation based federated learning method by utilizing unlabeled data on the server. A global model, including a feature extractor and classifier, is trained to approximate entropy-weighted pseudo-label on the server. Further, a self-supervised learning framework is applied on top to learn generic feature representation better.
The method is suitable when the amount of data per client or the number of clients is scarce. Empirically, it beats strong baselines such as FedAVG and FedDF.

**Summary Of The Review:**

This paper proposed FedDS, which is a continuation and extension of FedDF. The empirical results are well presented. More comparisons to related SOTA results would be better though. Some theoretical support for the convergence of the proposed method is lacking.

---

> ### Author Response · Authors · 2022-11-19
> **Response to Reviewer fQQY**
>
> ### The entropy-induced uncertainty is at the client level, but could it be that a local model performs well on part of the unlabelled data only? Can this pseudo label be calculated on a datapoint level?
>
> Thanks for the great suggestion for further extensions. It was understood as the idea of measuring the uncertainty of each data point of the server and filtering out data points that all the clients are not sure about. To check the effectiveness of this idea, we implemented a simple data rejection step in our FedDS algorithm, i.e., filtering out the data point of the server for which the maximum value of entropy-weighted pseudo label does not exceed a certain threshold. For the CIFAR-10 classification task, this additional step was shown to slightly improve the accuracy. It would be a great future work to explore more effective data-rejection methods.
>
>
> ### The choice of $\gamma$ (balance parameter for SSL loss) is not well justified
>
> For recap, parameter $\gamma$ is introduced for balancing the entropy-weighted ensemble distillation (EED) and the self-supervised learning (SSL) losses. In Section A.1.2 of the revised paper, we justified the choice of  $\gamma$ more clearly.
>
> The underlying reason why such a large $\gamma=300$ was applied is because the classification loss is implemented in a way that all the batch losses add up, instead of averaging. If we take into account such an accumulation effect, the equivalent balance parameter was in fact 4.68 = 300/64 (64 is the server’s batch size). For the chosen $\gamma$, the proportion of the rotation loss is about twice that of the classification loss at the 50th round for the i.i.d. setting in Fig. 4.
>
> We have run some manageable experiments to find the optimal value of $\gamma$, and found that as long as $\gamma$ is set in a way that the scales of the EED and the SSL losses do not differ too much, a small change does not seem to have a significant impact on the performance. In practice, one can use hyperparameter searching tools for better performance.
>
>
> ### The claim when $N_c≤N_s$, setting $B_{AVG} = false$ would give higher test accuracy is not well justified: when $N_c=N_s$, how could $B_{AVG}$ affect test accuracy? Would be nice to show it empirically.
>
> The numbers in green color in Tables 2 and 3 show the performance difference of FedDS by activating $B_{AVG}$.  In particular, Table 3 shows the experimental results by changing only Nc while fixing other conditions, and it shows that it’s better to deactivate $B_{AVG}$ when $N_c$ is smaller than $N_s$. When $N_c=N_s$, i.e., the case the reviewer mentioned, the third experiment in Table 3 shows that the gap due to activating $B_{AVG}$ is negligible (0.46%).
>
>
> ### How the train-test split is done on each client? Do they have the same distribution? This needs to be detailed about.
>
> A train-test split is not performed on each client. After first dividing the data into a training set and a test set, each client takes a part of the training set and uses all of them as a training set. The test set in our experiment is the test set provided by default in each dataset and it is used to measure the accuracy of the trained server model.

---

### Author Response · Authors · 2022-11-19
**Response to all reviewers**

We greatly thank all the reviewers (Reviewers fQQY, qQ7S, CynN) for their valuable comments and constructive suggestions. We believe that these comments do help us to substantially improve the quality of our paper. In particular, we were encouraged to hear from reviewers that our paper is well-motivated (Reviewers qQ7S, CynN), the proposed method is simple but effective (Reviewers qQ7S, CynN), and ablation study is rich and instructive (Reviewer fQQY).

In accordance with the reviewers’ comments, we have carefully selected and run additional experiments, derived a theoretical bound, and reported these results in the paper. The following is the summary of major changes:

- **Theoretical part:** (Reviewer qQ7S) We have added a generalization error bound analysis (Theorem 1) in Section A.3. We believe this allows to improve our understanding of our algorithm’s behavior.
- **Experimental part:**
1. (Reviewers qQ7S and CynN) We additionally investigated the effectiveness of our method in a normal FL situation with 100 clients in Section A.2.3. The results show that FedDS still outperforms other baselines.
2. (Reviewer CynN) We devised a baseline weighting method for measuring the uncertainty of clients’ prediction and compared it with our entropy-based weighting in Fig. 3. The comparison shows that it is not trivial to devise a weighting method that assigns higher weights to more correct predictions.
3. (Reviewer CynN) To see the effectiveness of our method under heterogeneous data setting, FedDS was evaluated with other baselines in Fig. 6 by using CIFAR-100 dataset for clients’ labeled dataset and TinyImageNet and ImageNet datasets for server’s unlabeled dataset. Fig. 6 shows that FedDS outperforms both FedDF and FedAVG in such heterogeneous data settings.
4. (Reviewer qQ7S) We evaluated FedProx as another baseline for the same settings as in Fig. 4. Its performance is between FedAVG and FedDF for all settings of Fig. 4.
5. (Reviewer CynN) We performed additional experiments for a larger dataset, ImageNet classification task, and reported the result in Section A.2.4 of the revised paper. Fig. 8 shows that our FedDS outperforms other baseline algorithms for such a larger dataset.
6. (Reviewer CynN) To see the effectiveness of our method when class distributions at the clients and at the server are different, we evaluated FedDS with other baselines in Fig. 9 for CIFAR-10 classification task with i.i.d. class distribution for clients and Dir(0.5) distribution for the server. We can see that our FedDS outperforms the baselines in such scenarios.
7. (Reviewer fQQY) In Section A.1.2 of the revised paper, we justified the choice of balance parameter $\gamma$ more clearly.

---

### Author Response · Authors · 2022-11-29
**Additional response to all reviewers**

Dear reviewers.

We have addressed all the reviewers' comments through this revision and rebuttal. If there is something that requires additional response, it would be appreciated if you let us know earlier so that we can respond within the discussion period.

---

### Decision · Program_Chairs · 2023-01-20

**Decision:**

Reject

**Justification For Why Not Higher Score:**

All 5 reviewers have consensus on enough concerns

**Justification For Why Not Lower Score:**

N/A

**Metareview: Summary, Strengths And Weaknesses:**

The paper studies federated learning using knowledge by utilizing unlabelled data on the server.

Unfortunately consensus among the reviewers remained that the paper in its current form remains below the bar even after the discussion phase, mainly due to the level of novelty.

We hope the detailed feedback helps to strengthen the paper for a future occasion.